# Chemical Composition and Biological Activity of *Tanacetum balsamita* Essential Oils Obtained from Different Plant Organs

**DOI:** 10.3390/plants11243474

**Published:** 2022-12-12

**Authors:** Milena D. Vukic, Nenad L. Vukovic, Ana D. Obradovic, Lucia Galovičová, Natália Čmiková, Miroslava Kačániová, Milos M. Matic

**Affiliations:** 1Department of Chemistry, Faculty of Science, University of Kragujevac, 34000 Kragujevac, Serbia; 2Department of Biology and Ecology, Faculty of Science, University of Kragujevac, 34000 Kragujevac, Serbia; 3Institute of Horticulture, Faculty of Horticulture and Landscape Engineering, Slovak University of Agriculture, Tr. A. Hlinku 2, 94976 Nitra, Slovakia; 4Department of Bioenergy, Food Technology and Microbiology, Institute of Food Technology and Nutrition, University of Rzeszow, 4 Zelwerowicza Str., 35-601 Rzeszow, Poland

**Keywords:** costmary, essential oils, chemical composition, antimicrobial activity, cancer cell lines, antitumor potential, redox homeostasis, cell migration

## Abstract

The aim of this study is to evaluate the chemical composition of *Tanacetum balsamita* L. essential oils (EOs) obtained from different plant organs, flowers (FEO), leaves (LEO), and stems (SEO), as well as to assess their biological properties. The results obtained by using GC and GC/MS analysis indicate that this plant belongs to the carvone chemotype. Moreover, we examined the oil’s antimicrobial and antitumor potential. Antimicrobial effects were determined using minimum inhibitory concentrations assay and the vapor phase method. Obtained results indicate better antimicrobial activity of investigated EO samples compared to the commercially available antibiotics. On the treatment with FEO, *Y. enterocolitica* and *H. influenzae* showed high sensitivity, while treatment with LEO and SEO showed the highest effects against *S. aureus*. The vapor phase method, as an in situ antibacterial analysis, was performed using LEO. Obtained results showed that this EO has significant activity toward *S. pneumoniae* in the apple and carrot models, *L. monocytogenes* in the pear model, and *Y. enterocolitica* in the white radish model. The potential antitumor mechanisms of FEO, LEO, and SEO were determined by the means of cell viability, redox potential, and migratory capacity in the MDA-MB-231 and MDA-MB-468 cell lines. The results show that these EOs exert antiviability potential in a time- and dose-dependent manner. Moreover, treatments with these EOs decreased the levels of superoxide anion radical and increased the levels of nitric oxide in both tested cell lines. The results regarding total and reduced glutathione revealed, overall, an increase in the levels of total glutathione and a decrease in the levels of reduced glutathione, indicating strong antioxidative potential in tested cancer cells in response to the prooxidative effects of the tested EOs. The tested EOs also exerted a drop in migratory capacity, which indicates that they can be potentially used as chemotherapeutic agents.

## 1. Introduction

Antibiotic resistance, alongside cancer as one of the main causes of mortality, has become a wide field for research of novel therapeutic principles. Medicinal and aromatic plants are known to have a broad spectrum of biologically active secondary metabolites that humans have used since ancient times. Either alone, as pure compounds, or in synergy as part of plant extracts or essential oils, naturally occurring secondary metabolites represent a unique source of compounds with significant therapeutic potential that has been extensively studied in recent years.

Essential oils (EOs) have attracted great attention in the scientific community, first as this plant product is recognized as safe by the US FDA (Food and Drug Administration) and the EPA (Environmental Protection Agency) and also as they are widely used in the food, pharmaceutical, and cosmetic industries. These volatile liquids, or semi-liquids, extracted from the plant, are characterized by their broad spectrum of biological functions as antioxidant, antimicrobial, anti-allergic, antiviral, enzyme inhibitory, insecticidal, anti-tumor, and pro-apoptotic [1,2,3,4,5].

Lately, costmary (*Tanacetum balsamita* L.) has once again drawn the scientific community’s attention because of its diverse biological activities. *Tanacetum balsamita*, belonging to the Asteraceae family and the *Tanacetum* genus, has many different documented scientific names, but it is popularly known as costmary [6,7,8,9,10]. This genus contains about 200 species distributed over Asia, Africa, Europe, and the America [11,12,13,14]. Since ancient times, the *Tanacetum* genus has been used in traditional medicine in the treatment of different disorders (migraine, stomach aches, toothaches, infertility, psoriasis, allergies, nausea, menstrual problem, in the treatment of inflammation wounds and ulcers, etc.) [6,10,13,15,16]. *T. balsamita* was first mentioned during the early modern period, in the 18th century, and classified as a laxative and astringent, good for stomach aches and in conditions of melancholy and hysteria. Traditionally, costmary has been used as a spice, herbal tea, beer flavoring, and salad [7,10].

Phytochemical investigation of this species led to the identification of volatile oil, phenylpropane derivatives, flavonoids, sesquiterpene lactones, tannins, and oligo-elements [17]. It is well-known that chemical polymorphism influences the volatile composition of essential oils. Considering predominant terpenes, four chemotypes were noted for *T. balsamita*: carvone type, camphor type, camphor-thujone type, and carvone-α-thujone type [7,10,16,18]. Carvone has been characterized as the main component in the essential oils of many species of the Lamiaceae and Asteraceae families. This monoterpene ketone has been shown to have promising pharmacological properties, such as neuroprotective, antidiabetic, antifungal, antibacterial, antibiofilm, and anticancer effects, which qualified it as a key candidate in drug development [19].

Cancer is one of the most persistent diseases with high mortality worldwide. The efficiency of established chemotherapies has been reduced because of the emergence of chemoresistant cancer phenotypes, emphasizing the need for novel therapeutic combinations with higher cytotoxicity against malignant cells and minimal deteriorating outcomes in healthy tissues. Breast cancer is the second leading cause of cancer mortality in females, resulting in more than half a million deaths each year. Naturally occurring plant compounds such as terpenoids, phenolics, flavonoids, and alkaloids exert significant antitumor potential [20].

Reactive oxygen and nitrogen species (ROS and RNS) may affect carcinogenesis and tumor progression by genetic and epigenetic pathways and the disturbance in redox homeostasis has been detected in various tumors, including breast cancer [21].

Based on the above facts, the aim of this study was to evaluate the chemical composition (chemotype) and biological effects of *T. balsamita* essential oils obtained from flowers (FEO), leaves (LEO), and stems (SEO). Chemotype was determined using GC and GC-MS analysis; antibacterial activity was examined on three Gram-positive (G^+^) and three Gram-negative (G^−^) bacterial strains. Using the vapor phase method, EO obtained from leaves was further analyzed on selected food models (apple, pear, carrot, and white radish) to consider its potential application as a natural agent in preservation. Antitumor potential, effects on redox status, and migratory capacity on two human breast cancer cell lines, MDA-MB-231 and MDA-MB-468, of the obtained Eos, FEO, LEO, and SEO, was determined to establish the potential mechanism of their antiproliferative properties. In addition, to examine the biocompatibility of the tested EOs, we also evaluated their cytotoxic effects on healthy human lung fibroblast (MRC-5).

## 2. Results

### 2.1. Volatile Composition of Examined Essential Oils

The identified volatiles in the *T. balsamita* FEO, LEO, and SEO, the percentages of their contents, and their calculated and literature retention indices are presented in Table 1, while Table 2 and Table 3 show the percentage amounts of each class of identified compounds and the number of compounds identified in the corresponding classes, respectively.

Overall, in all tested samples, 103 compounds were identified, out of which 91 compounds was presented in FEO, 75 in LEO, and 74 in SEO (Table 1). As can be seen from Table 2 and Table 3, in all examined Eos, the dominant class of compounds were found to be oxygenated monoterpenes, whose content percentages varied from 90.5% in FEO (31 compounds) to 74.9% in SEO (27 compounds). Carvone (monoterpene ketone) was the main component in all EOs obtained from different plant organs, identified in amounts of 54.2%, 52.1%, and 47.7% (FEO, LEO, and SEO, respectively).

Alongside carvone, the chemical composition of FEO was also accompanied by high amounts of monoterpene ketones (71.6%, 11 compounds), trans-dihydrocarvone (7.7%), and β-thujone (6.4%). The second class of identified compounds in high amount belonging to this plant organ are monoterpene alcohols (16.4%, 16 compounds), out of which trans-*p*-mentha-1(7),8-dien-2-ol (3.5%), cis-*p*-mentha-1(7),8-dien-2-ol (2.8%), and cis-carveol (2.2%) were found to be the dominant ones. Other volatiles presented in FEO were observed in quantities lower than 2.2%.

In contrast to FEO, in the LEO sample, besides carvone, from the class of monoterpene ketones (69.1%, 11 compounds) α-thujone with a contribution of 11.4% was presented in a high amount. Interestingly, in the plant’s flower organ, α-thujone was found in a considerably lower amount of 1.0%. However, the quantity of β-thujone (1.5%), trans-dihydrocarvone (0.7%), and cis-carveol (0.5%) significantly decreased in LEO compared to in the FEO sample. The LEO sample was also characterized by monoterpene alcohols in the content percentage of 13.7% (11 compounds), with trans-*p*-mentha-1(7),8-dien-2-ol (3.6%) and cis-*p*-mentha-1(7),8-dien-2-ol (2.7%) as the dominant ones. Compared to FEO, in the LEO sample, around three times the amount of monoterpene ether 1,8-cineole (5.9%) and sesquiterpene hydrocarbon β-bisabolene (4.0%) was noticed. Other constituents belonging to the above-mentioned class of compounds identified in leaves of *T. balsamita* alongside volatiles from the class of nonterpenic compounds, phenylpropanoids, and sesquiterpenes were quantified in amounts equal to or below 1.5%.

If we compare the composition of the EOs obtained from different plant organs, it can be noticed that the quantity of the main compound carvone is 6.5% and 4.4% lower in stems than in flowers and leaves, respectively. As in previous samples, oxygenated monoterpenes, with 27 identified compounds representing 74.9% of the total oil composition, are the dominant class of volatiles presented in SEO. Like in LEO, in the SEO sample, α-thujone was found to be the second major compound (7.8%), and a decrease in the amounts of β-thujone (1.2%), trans-dihydrocarvone (1.0%), and cis-carveol (1.3%) was notable, compared to the composition of FEO. Monoterpene alcohols, trans-*p*-mentha-1(7),8-dien-2-ol (3.4%), and cis-*p*-mentha-1(7),8-dien-2-ol (2.5%) were also presented in high amounts as in FEO and LEO. Results presented in Table 1 also indicate that the concentration of 1,8-cineole (1.8%) in SEO is similar to its concentration in FEO and considerably lower than in LEO. However, the quantity of β-bisabolene (7.7%) is almost two times higher in SEO compared to LEO. In addition, it is significant to mention that the amount of τ-muurolol (1.2%) in stems is three times higher compared to its amount in other plant organs. The quantities of other identified compounds in SEO are lower than 1.2%.

### 2.2. Antibacterial Activity of Tested Essential Oils

Obtained EOs from flowers, leaves, and stems of *T. balsamita*, as well as two standard antibiotics, were evaluated for activity on three G^+^ and three G^−^ bacterial strains, and results are presented in Table 4 as MIC50 and MIC90 values in μg/mL. Recorded MIC50 and MIC90 values were in the range of 0.246 ± 0.009 μg/mL and 0.61 ± 0.011 μg/mL, respectively, out of which we can generally conclude that the tested EOs have good activity toward both, G^+^ and G^−^ acteria.

Compared to standard antibiotics meropenem and vancomycin, FEO demonstrated better activity toward G^−^
*Y. enterocolitica* (with MIC50 0.016 ± 0.002 μg/mL and MIC90 0.022 ± 0.003 μg/mL) and *H. influenzae* (with MIC50 0.063 ± 0.004 μg/mL and MIC90 0.088 ± 0.002 μg/mL), while on *E. coli* this EO showed stronger antibacterial effectiveness only compared to meropenem (with MIC50 0.246 ± 0.003 μg/mL and MIC90 0.436 ± 0.001 μg/mL). Considering G^+^ bacteria strains, this EO showed high activity (higher than tested standards) toward *L. monocytogenes* with MIC50 of 0.246 ± 0.003 μg/mL and MIC90 of 0.436 ± 0.001 μg/mL, while *S. aureus* and *S. pneumoniae* were more resistant on the treatment with FEO.

The LEO and SEO showed very high antibacterial effectiveness against the growth of all tested G^+^ bacteria strains compared to tested standards. The highest sensitivity on treatment with LEO and SEO (MIC50 0.016 ± 0.003 μg/mL and MIC90 0.022 ± 0.001 μg/mL, and MIC50 0.009 ± 0.002 μg/mL and MIC90 0.011 ± 0.001 μg/mL, respectively) was observed for *S. aureus*. Concerning the G^−^ bacterial strain, the most resistant to treatment with leaves and stems EOs was *E. coli*. The strongest effect these two EOs demonstrated was toward G^−^
*Y. enterocolitica*, even in comparison with standard antibiotics (for LEO MIC50 0.036 ± 0.002 μg/mL and MIC90 0.054 ± 0.004 μg/mL, and SEO MIC50 0.036 ± 0.002 μg/mL and MIC90 0.054 ± 0.004 μg/mL).

### 2.3. In Situ Antibacterial Analysis on a Food Model

To further evaluate the antimicrobial potential of EO obtained from leaves of *T. balsamita* (LEO), we performed in situ antibacterial analysis on apple, pear, carrot, and white radish as food models on which we have grown the same bacterial strains as used in the evaluation of MIC50 and MIC90. The obtained results are presented in Table 5.

The results in Table 5 of the in situ evaluation of leaves EO revealed moderate antibacterial activity in all concentrations applied to the growth of G^+^
*S. pneumoniae* on apple, with the lowest concentration applied (3.9 μL/L) showing the strongest inhibitory effect (35.25 ± 1.07%). The highest concentration of tested EO had similar effectiveness in the inhibition of *S. aureus* (44.23 ± 0.99%). Against *L. monocytogenes*, inhibitory action of LEO was noted only in the applied concentration of 7.8 μL/L (25.30 ± 1.06%), while in other concentrations, probacterial activity was displayed. Considering G^−^ bacterial strains, the tested EO showed moderate antibacterial activity in the highest concentration applied to the growth of *E. coli* (34.35 ± 0.98%) and in the lowest concentration against *H. influenzae* (23.45 ± 0.90%) and *Y. enterocolitica* (6.73 ± 1.20%) growing on apple.

The in situ evaluation of the G^+^ and G^−^ bacteria growing on pears generally showed moderate inhibitory activity of LEO. The increase in growth of bacteria was observed for G^+^
*S. aureus* in treatment with LEO at the concentrations of 3.9 μL/L and 7.8 μL/L and G^−^
*H. influenzae* at the concentration of 7.8 μL/L. Against G^+^
*L. monocytogenes* and *S. pneumoniae* growing on pears, the tested EO showed the strongest inhibitory potential in the lowest concentrations applied (36.32 ± 1.12% and 25.24 ± 1.04%, respectively). Considering the inhibition of G^−^ bacterial strains, strong effects of LEO were observed for *H. influenzae* growing on pears in treatment with the lowest concentration applied (35.31 ± 0.92%) and *Y. enterocolitica* in the highest concentration applied (34.22 ± 2.02%).

Antibacterial activity of the vapor phase of LEO on bacteria growing on carrots revealed low to moderate effects. For the examined G^+^ bacteria strains, the tested EO showed low antibacterial effectiveness. For *L. monocytogenes*, an increase in bacterial growth with an increase in applied concentration of LEO was observed. However, treatment with 15.6 μL/L of LEO showed the strongest growth inhibition of *S. pneumoniae* (17.43 ± 1.00%), and in the concentration of 3.9 μL/L, the inhibition of *S. aureus* (19.32 ± 0.95%) was notable. The most sensitive on treatment with LEO in the lowest concentration applied was G^−^
*H. influenzae* (47.87 ± 1.35%), while in the highest applied concentration, probacterial activity (−45.25 ± 1.73%) was notable. Inhibition of *E. coli* growing on carrots was most efficient in treatment with 7.8 μL/L of LEO (25.06 ± 0.45%), and of *Y. enterocolitica* in treatment with the concentration of 3.9 μL/L (13.31 ± 0.95%).

Out of the G^+^ bacterial strains growing on white radish, the most sensitive on treatment with LEO was *S. pneumoniae* with a growth inhibition rate of 35.29 ± 1.06%, followed by *S. aureus* where low inhibition of growth with a rate of 17.67 ± 0.48% (in the applied concentration of LEO of 15.6 μL/L for both) was observed. All other treatments of G^+^ bacterial strains growing on white radish showed probacterial activity. On the contrary, for G^−^ bacteria, a significant antibacterial activity of LEO was observed. *E. coli* was inhibited at a concentration of 15.6 μL/L with an inhibitory effect of 87.35 ± 1.97%, while the growth of *Y. enterocolitica* showed the strongest inhibition rate at treatment with 3.9 μL/L (87.09 ± 1.48%), and *H. influenzae* was the most effectively inhibited by the vapor phase of LEO at 7.8 μL/L (74.23 ± 1.04%).

### 2.4. Determination of Cell Viability (MTT Assay)

To assess the selectivity of the tested EOs toward cancer cells, we examined the effects on the viability of normal human lung fibroblast cell line MRC-5 of all applied concentrations of the oils (Figure 1). The obtained results suggest that the EOs exerted multifold stronger effects on the inhibition of viability of human breast cancer cells compared to MRC-5 cells, especially in the two highest concentrations (100 µg/mL and 200 µg/mL) with SEO, suggesting that these oils exhibit desirable biocompatibility for potential further usage against cancer cells.

With the aim of determining the antiproliferative effects of extracted essential oils FEO, LEO, and SEO, an MTT cell viability assay was conducted on two human breast cancer (MDA-MB-231 and MDA-MB-468) cell lines. Results obtained after 24 h and 72 h of incubation with various concentrations of *T. balsamita* EOs obtained from three different organs are presented in Figure 2. Generally, the results indicate significant antiproliferative effects compared to the nontreated cells, and dose-dependent inhibition of cell viability was observed for both cell lines after treatment with FEO, LEO, and SEO.

### 2.5. The Effects of T. balsamita Essential Oils on Redox Status in Tumor Cells

The effects of short-term and long-term exposure of human breast MDA-MB-231 and MDA-MB-468 cells to different concentrations of treatment from *T. balsamita* on redox status parameters were monitored. The obtained results so far indicate that these EOs decrease the viability of tumor MDA-MB-231 and MDA-MB-468 cell lines and have a much weaker effect on the viability of human fibroblast cells (MRC-5). To assess the contribution of oxidative stress to the observed antiproliferative activity, we also examined the effects of FEO, LEO, and SEO on oxidative stress markers, precisely on the production of superoxide anion radical (O_2_^•−^), nitrites (NO_2_^−^), and total and reduced glutathione, as established indicators of oxidative stress and the influence of the examined treatments on the redox homeostasis of breast cancer.

#### 2.5.1. Determination of Superoxide Anion Radical (NBT Assay) and Nitrites (Griess Assay)

In Figure 3, the estimated levels of superoxide anion radical (O_2_^•−^) in MDA-MB-231 and MDA-MB-468 cells after 24 h and 72 h of incubation with various concentrations of FEO, LEO, and SEO are presented. Compared to the control cells, all applied concentrations exhibited a significant reduction in O_2_^•−^ levels in the tested tumor cells at both time treatments. Overall, the strongest drop of O_2_^•−^ production was exhibited by SEO on the MDA-MB-231 cell line.

Nitric oxide (NO) is an important signaling molecule in numerous physiological and pathological conditions. Therefore, we have evaluated the production of nitrites in the MDA-MB-231 and MDA-MB-468 cell lines after 24 h and 72 h of incubation with FEO, LEO, and SEO. The obtained results are presented in Figure 3. Treatments with all three EOs showed a significant increase in the production of nitrite by both cell lines compared to the control. Since NO is a potent signaling molecule, the increase could affect various metabolic pathways.

#### 2.5.2. Determination of Total and Reduced Glutathione

The data presented in Figure 4 show the effects of the investigated treatments on total and reduced glutathione levels (GSH) in MDA-MB-231 and MDA-MB-468 cells.

The concentration of total glutathione after 24 h and 72 h of incubation with various concentrations of all three EOs showed an increase in the levels of glutathione compared to the control. The level of total glutathione after short-term (24 h) exposure in MDA-MB-468 cells in all applied concentrations of all three EOs was significantly increased compared to control cells. The increase in total glutathione level was dose dependent. The tested EOs exerted dose-dependent reduction in reduced glutathione (GSH) levels in both breast cancer cell lines, with the maximal reduction at the highest applied concentration of 200 µg/mL. After 72 h of the incubation period, the same dose-dependent trend was observed in the levels of GSH.

### 2.6. Transwell Assay for Cell Migration

Cell migration capacity represents one of the most prominent parameters in monitoring tumor progression, so an efficient antitumor agent should affect cell mobility. To examine the effects of FEO, LEO, and SEO treatments on the migration capacity of cancer cells, a 2D transwell migration assay was performed. The results indicate a significant dose-dependent decrease in the cell migration index of both MDA-MB-231 and MDA-MB-468 cells exposed to these EOs compared to the nontreated cells, as presented in Figure 5.

## 3. Discussion

Reports made until now, considering the chemical composition of essential oils of *T. balsamita*, have revealed a total of about 200 compounds identified in aerial parts, as well as in flowers, leaves, and stems, that were investigated separately and at different growth phases [7,10,13,22,23,24]. The interesting element is the differences in all of these reports. For example, Bagci et al. reported that trans-chrysanthenol was the major compound of the aerial part of plant-derived oil, followed by chrysanthenyl acetate and linalool oxide [22]. Some other authors reported carvone, β-thujone, 1,8-cineole, and α-thujone to be the main volatiles of this species’ EOs [13,14,16,24,25]. In addition, Jaimand and Rezaee reported bornyl acetate and pinocarvone as the main components of subspecies balsamitoides [23]. The literature data also reveal numerous EO chemotypes that are in close relationship with the geographical origin, the vegetative period, or the method used to extract the EOs.

In our study, all analyzed oil samples are characterized by a high amount of monoterpene ketone carvone, which was most abundant in essential oil derived from flowers. In addition, the concentrations of β-thujone, trans-dihydrocarvone, and cis-carveol were found to be significantly higher in flower oil compared to oils obtained from leaves and stems. The EO obtained from leaves is characterized by slightly higher amounts of 1,8-cineole and α-thujone in comparison to oils obtained from other plant organs, while in the EO obtained from stems, the concentration of β-bisabolene was highest. Comparing our results with the ones already published, it can be concluded that the investigated *T. balsamita* matches the carvone chemotype. Concerning the abundance of compounds class, FEO and LEO are characterized by a higher amount of oxygenated monoterpenes compared to SEO. However, EO obtained from stems is characterized by higher amounts of sesquiterpenes in comparison to ones obtained from flowers and leaves.

Previous reports showed that different *Tanacetum* species’ essential oils have proven their antibacterial activities [26]. The antibacterial activity of *T. balsamita* was also recorded and characterized as moderate to strong. The differences between the published results may be attributed to the variety of the chemical profiles of this species [10,13,14,22]. As carvone is the main constituent of all investigated EOs, as presented by the results in this paper, it can be responsible for their displayed strong effects [19]. Previous work by Yousefzadia et al. (in which the chemical profile of the obtained EO is similar to the ones described in this paper) demonstrated that essential oil obtained from the aerial parts of *T. balsamita* showed higher effectiveness toward the G^+^ compared to the G^−^ bacterial strains, whereas our results showed no significant differences in inhibition between these two types of bacterial strains. Obviously, we cannot overlook the synergistic effect of the other components present in minor amounts that may affect the effectiveness of the exhibited antimicrobial effect [14]. Overall, we can conclude that our results indicate better antimicrobial activity toward selected bacterial strains of investigated EO samples compared to the commercially available antibiotics meropenem and vancomycin. These results indicate that the costmary herb may be considered a promising product for use in the pharmaceutical and food industries.

It is known that fruits and vegetables are characterized by a short shelf life because of weight loss and decay, caused mainly by fungal activity. This is a huge problem for producers, stakeholders, and consumers [27]. Therefore, in recent years, there has been a significant increase in research work on the postharvest control of phytopathogens through alternative natural processes, such as the use of EOs from aromatic plants [28].

The effect of the vapor phase of LEO against G^+^ and G^−^ was recorded using in situ analysis on apples, pears, carrots, and white radishes. The *T. balsamita* EO obtained from leaves was most effective against *S. pneumoniae* in the apple model, *L. monocytogenes* in the pear model, *S. pneumonia* in the carrot model, and against *Y. enterocolitica* in the white radish model. No other authors focused on the antimicrobial activity of the volatile vapor of this EO considering that these volatile liquids have been already identified as natural food additives that can find useful applications in food preservation. Moreover, they are functional alternatives to synthetic chemicals in food preservation because of their good efficacy and because they are environmentally friendly [2].

The majority of antitumor agents affect cancer cell viability. In our study, all three EOs exerted significant potential in reducing breast cancer cell viability, as shown by the MTT assay. The antiproliferative activity of FEO, LEO, and SEO was the strongest at the concentration of 100 µg/mL and 200 µg/mL after both incubation times. By comparing the obtained results, we conclude that treatments with these EOs on both cell lines, MDA-MB-231 and MDA-MB-468, were time and dose dependence. The treatment with SEO after 72 h of incubation in the highest applied concentration was the strongest (66.2%) in a concentration of 200 µg/mL on the MDA-MB-231 cell line. However, the values of the antiproliferative effects do not exceed 60% compared to control cells, except for the highest concentration of 200 µg/mL.

Cancer cells exhibit disturbances in redox homeostasis and, often, the presence of oxidative stress is a hallmark of these cells. ROS appear to be involved in the regulation of various physiological pathways, including signal transduction, apoptosis, and differentiation. Recently, emerging evidence has suggested the involvement of ROS and the aberrant activation of redox-sensitive signaling pathways in tumor invasion and migration. Some ROS-regulated proteins play key roles in tumor metastasis, including the effects on integrins and matrix metalloproteinases [29]. In both breast cancer cell lines, the decrease in O_2_^•−^ concentrations of the tested oils were recorded, with slight differences in the trends between the cell types. The treatments of MDA-MB-231 cells exerted the strongest drop in O_2_^•−^ levels at the lowest applied concentrations, while in MDA-MB-468 cells the strongest decrease in the O_2_^•−^ level was recorded at the highest applied concentration of EOs. Some biotherapeutics may enhance the effects of cytotoxic regimes by altering redox homeostasis and improving the response rate of tumors to chemotherapeutic agents, while some others can ameliorate their antitumor activity [30,31]. Moreover, the obtained results indicate that the exerted antioxidant impact could be an important pathway in the regulation of cancer cell progression and viability. Since the treatments induce a decrease in reduced glutathione levels, they could be the basis of the recorded drop in the O_2_^•−^ level in the study. Therefore, we suggest that the tested EOs exerted pro-oxidative effects, followed by the strong antioxidative response of both tested breast cancer cell lines used in the study. Overall, the investigated EOs exerted an inhibitory impact on human breast cancer cell viability in correlation with a decreased O_2_^•−^ level.

NO is reported to have antitumor activities, as well as protumor properties, depending on the timing, concentration, and tissue type [32]. The changes in the production of NO could affect various signaling pathways that involve nitric oxide, leading to potential antitumor outcomes. Since NO has a half-life of only several seconds in a solution rich in superoxide anion radical, in surroundings with a low level of superoxide anion radical NO has much greater stability and prolonged signaling effects [33,34]. Accordingly, in both cell lines, all applied concentrations at both time treatments induced the increase in NO production compared to control, especially in the lowest concentration, which correlates with the strongest reduction in O_2_^•−^ concentrations in these cells. The obtained data indicate that stimulation in NO production and/or bioavailability significantly contributes to the recorded antitumor activity of the tested EOs. The obtained data indicate that the tested compounds are suitable for further investigations in designing novel approaches for antitumor therapy.

Reduced glutathione, a tripeptide consisting of cysteine, glutamate, and glycine, is one of the strongest antioxidative components in cells, maintaining intracellular thiol status and detoxicating various metabolites, essential for optimal activity of some enzymes and other cellular macromolecules [35]. Decreased levels of reduced glutathione suggest a considerable oxidative burst triggered by the tested EOS and the strong antioxidative potential in cancer cells, which is their common hallmark. In addition, the increase in total glutathione indicates significant de novo synthesis triggered by the treatments, suggesting the peculiar intense potential of breast cancer cells to augment the new capacities of glutathione protection against oxidative insult [36]. However, despite the considerable antioxidative protective potential of these cells, the tested EOs significantly reduced their viability/proliferation.

All three examined EOs exerted a drop in cell migration capacity, indicating their favorable usage as chemotherapeutic agents. Both short-term and long-term exposure decreased migration capacity compared to control. The strongest decrease in migration capacity (42.1% compared to the control) was provoked by long-term exposure (72 h) in a concentration of 10 µg/mL of FEO on MDA-MB-468 cells. Since numerous studies indicate that nitric oxide can inhibit cell migration, based on our results regarding ROS production we can suggest that the elevated nitric oxide bioavailability caused by the drop in O_2_^•−^ leads to the antimigratory outcomes recorded in this study.

## 4. Materials and Methods

### 4.1. Plant Material

The plant material of *Tanacetum balsamita* was obtained from Ovčar-Kablar Gorge (GPS 43.907004 N 20.3327868 E, Serbia, September 2020). The voucher sample is deposited at the Herbarium of the Institute of Botany and Botanical Garden “Jevremovac”, University of Belgrade (voucher number No 17696). Carefully separated plant materials, flowers (FEO), leaves (LEO), and stems (SEO) were air dried at room temperature in darkness.

### 4.2. Isolation of Essential Oils

Dried flower (FEO), leaf (LEO), and stem (SEO) parts of the plant were subjected to hydro-distillation in a Clevenger-type apparatus. Hydro-distillation was performed for a period of 4 h. The obtained EOs were dried over anhydrous sodium sulfate. Until further use, EOs were stored in sealed vials in the dark at 4 °C.

### 4.3. Gas Chromatography and Gas Chromatography-Mass Spectrometry Analysis

To determine the volatile composition of the *T. balsamita* essential oils obtained from different plant organs, GC and GC-MS analyses were performed. For that purpose, an Agilent Technologies (Palo Alto, Santa Clara, CA, USA) 6890N gas chromatograph was employed. The chromatograph was equipped with an HP-5MS capillary column (30 m × 0.25 mm × 0.25 µm) and interfaced with a quadrupole mass spectrometer 5975B (Agilent Technologies, Santa Clara, CA, USA) that is operated by HP Enhanced ChemStation software (Agilent Technologies).

The chromatographic conditions of GC and GC-MS analysis were as follows: the temperature program was 50 °C to 90 °C (increasing rate, 5 °C/min), 90 °C to 120 °C (increasing rate, 4 °C/min), hold 3 min at 90 °C, and 120 °C to 290 °C (increasing rate, 5 °C/min), and hold 15 min at 120 °C; total run time was 67 min; injection volume was 1 µL (the EO samples were diluted in hexane, 10% solution); split ratio was 40.8:1; helium 5.0 was the carrier gas with a flow rate of 1 mL/min; split/splitless injector temperature was set at 280 °C; MS source temperature was 230 °C; MS quadruple temperature was 150 °C; mass scan range was 35–550 amu at 70 eV; and solvent delay time was 3.2 min for oil sample analysis, while in the case of n-alkanes (C7–C35), the solvent delay time was 2.1 min to obtain the retention index for n-heptane (identified at 2.6 min).

The identification of the volatile constituents was performed in comparison of their retention indices (RI) with retention indices of the n-alkanes (C7–C35) series [37,38]. Likewise, compounds were identified by comparing their spectral data with the reference spectra reported in the literature and stored in the MS library (Wiley7Nist) that is merged with the HP Enhanced ChemStation software. Semi-quantification of each component was performed on GC-FID (using the same HP-5MS capillary column), considering amounts higher than 0.1%.

### 4.4. Antibacterial Activity

#### 4.4.1. Tested Microorganisms

Gram-positive bacteria (*Streptococcus pneumoniae* CCM 4501, *Listeria monocytogenes* CCM 4699, and *Staphylococcus aureus* subsp. *aureus* CCM 2261) and Gram-negative bacteria (*Escherichia coli* CCM 3954, *Yersinia enterocolitica* CCM 7204, and *Haemophilus influenzae* CCM 4456) were obtained from the Czech collection of micro-organisms (Brno, Czech Republic).

#### 4.4.2. Minimum Inhibitory Concentrations (MIC)

Bacterial inoculum was cultivated for 24 h in Mueller Hinton broth (MHB, Oxoid, Basingstoke, UK) at 37 °C. There was 50 μL of inoculum with an optical density of 0.5 of the McFarland standard added to a 96-well microtiter plate. Subsequently, the EOs were prepared by serial dilution to a concentration range of 1 μg/mL to 0.0002 μg/mL and antibiotics (meropenem and vancomycin) of 1 μg/mL to 0.005 μg/mL in MHB and 100 μL of suspension were thoroughly mixed with bacterial inoculum in the wells. Bacterial strains were incubated for 24 h at 37 °C. MHB with EOs was used as a negative control and MHB with inoculum was used as a positive control of the maximal growth. For nonadherent microorganisms, the absorbance was measured after the incubation period at 570 nm by Glomax spectrophotometer (Promega Inc., Madison, WI, USA). The absorbance was measured at 570 nm. The concentration of EOs whose absorbance was lower than the absorbance of the maximal growth control was determined as the minimum inhibitory concentration. The test was prepared in triplicate.

#### 4.4.3. In Situ Analysis on a Food Model

All six bacterial strains: Gram-positive bacteria (G^+^) (*S. pneumoniae*, *L. monocytogenes*, and *S. aureus*) and Gram-negative bacteria (G^−^) (*E. coli*, *Y. enterocolitica*, and *H. influenzae*) were used to estimate an in situ antimicrobial activity of the vapor phase of *T. balsamita* essential oil obtained from leaves (LEO). For the growth of microorganism species, the substrates used were commercially consumed food models—apple, pear, carrot, and white radish.

Warm MHA was poured into 60 mm Petri dishes (PD) and the lid. Sliced apples, pears, carrots, and white radish (0.5 mm) were placed on agar. Then an inoculum was prepared as previously described. The EO was diluted twice in ethyl acetate to 15.6, 7.8, and 3.9 μL/L and it was used for sterile filter paper inoculation. The filter paper was placed in it for 1 min to evaporate the remaining ethyl acetate, sealed, and incubated at 37 °C for 7 days. In situ bacterial growth was determined using stereological methods. In this concept, the volume density (vv) of bacterial colonies was first estimated using ImageJ software, counting the points of the stereological grid hitting the colonies (P) and those (p) falling to the reference space (growth substrate used). The volume density of bacterial colonies was consequently calculated as follows: vv (%) = P/p. The antibacterial activity of EO was defined as the percentage of bacterial growth inhibition (BGI):BGI = [(C − T)/C] × 100(1)
where C and T were bacterial growth (expressed as *v/v*) in the control group and the treatment group, respectively. The negative results represented growth stimulation.

### 4.5. Cell Cultivation and Treatment

The human lung normal fibroblast cell line, MRC-5, and two breast cancer cell lines, MDA-MB-231 and MDA-MB-468, were obtained from the American Tissue Culture Collection. These cells were propagated and maintained in DMEM and supplemented with 10% FBS and a combination of antibiotics (100 IU/mL penicillin and 100 µg/mL streptomycin). The cells were grown in 75 cm^2^ culture flasks and supplied with 15 mL DMEM at a confluence of 70% to 80%. The cells were seeded in a 96-well microplate (10,000 cells per well) and cultured in a humidified atmosphere with 5% CO_2_ at 37 °C. After 24 h of cell incubation, 100 μL of medium containing various doses of treatment (1 µg/mL to 200 µg/mL) was added to each well of the microplate, and the cells were incubated for 24 h and 72 h, after which the evaluation of cell viability, superoxide anion radical, nitrites, and glutathione were measured. Essential oils of *T. balsamita* were used in the experiments. The stock solution was prepared in the concentration of 10 mg/mL, while the following experimental concentrations were applied to the cells: 1, 10, 20, 50, 100, and 200 µg/mL. The stock solution was prepared by dissolving the essential oils in a DMEM/DMSO mixture in a ratio of 9:1 (*v/v*). All experimental concentrations were obtained by serial dilutions of the stock solution, so DMSO concentrations decreased continuously and never exceeded 0.2% in the maximal applied concentration. For migration capacity assessment, 500,000 cells per mL were seeded in a six-well plate and cells were treated with two concentrations 1 µg/mL and 10 µg/mL. Nontreated cells were used as control. All concentrations were tested in triplicate for all the methods.

### 4.6. MTT Assay

The viability of the cells was determined using an MTT assay [39]. Briefly, the cells were plated at a density of 100,000 cells/mL (100 µL/well) in 96-well plates with DMEM. After a period of incubation (24 h), at a temperature of 37 °C and 5% CO_2_, the six different concentrations of all essential oils in concentration (1 to 200 µg/mL) were dissolved in DMEM and were added to each well (100 µL/well). Treated and control cells (cultured only in a medium) were incubated for 24 and 72 h and afterward used to determine the cell viability by adding 20 µL of MTT (concentration of 5 mg/mL) to each well. After this reaction, the formed crystals were dissolved in 20 µL of DMSO. The color formed was measured on an ELISA reader at a wavelength of 550 nm. The percentage of viable cells was calculated as the ratio between the absorbance at each dose of the treatment and the absorbance of the nontreated control multiplied by 100 to get a percentage.

### 4.7. Determination of Superoxide Anion Radical (NBT Assay)

The concentration of superoxide anion radical (O_2_^•−^) was determined using the well-known spectrophotometric method [40]. This method is based on the reduction of nitroblue tetrazolium (NBT) to nitroblue formazan in the presence of O_2_^•−^. An assay was performed by adding 20 μL of 5 mg/mL NBT to each well, followed by cell incubation for 1 h at 37 °C in 5% CO_2_. To quantify the formazan production, formazan was solubilized in 20 μL DMSO. The absorbances were measured on an ELISA reader at 550 nm. The concentrations of O_2_^•−^ were expressed as nanomoles per milliliter (nmol O_2_^•−^/mL) in 10^5^ cells.

### 4.8. Determination of Nitrites (Griess Assay)

The spectrophotometric determination of nitrites (NO_2_^−^) as an indicator of the nitric oxide (NO) level was performed by using the Griess method [41]. Equal volumes of 0.1% (1 mg/mL) *N*-1-napthylethylenediamine dihydrochloride and 1% (10 mg/mL) sulfanilamide solution in 5% phosphoric acid were mixed to form the Griess reagent immediately before application to the plate. During the 10 min of incubation (at room temperature, protected from the light sources) the purple color was developed. After incubation, absorbances were measured on an ELISA reader at 550 nm and the nitrite concentration was expressed in μmol NO_2_^−^/mL in 10^5^ cells.

### 4.9. Reduced and Total Glutathione Concentration

After the initial incubation, the treatment was added and incubated for 24 h and 72 h, respectively. The determination of the reduced form of glutathione (GSH) is based on the interaction of GSH with sulfide reagent DTNB to form a yellow product, 5′-thio-2-nitrobenzoic acid (TNB) [42]. The treatment solution was removed and 150 μL of 2.5% (SSA) was added to each well. The cells were sonicated for 10 s and 50 μL of supernatant was poured from each well into a new plate. After that, 50 μL of the reaction mixture prepared before the analysis (1 mM DTNB dissolved in 100 mM phosphate buffer) was added [43]. For measurement of total glutathione, the reduction procedure of the oxidized portion of glutathione (GSSG) was performed by using 50 μL of the reaction mixture (1 mM DTNB, 1 mM NADPH, and 0.7 U glutathione reductase in 100 mM phosphate buffer) and added into 50 μL of supernatant [42]. The plates were incubated in the dark for 5 min at room temperature, and the absorbances were measured on an ELISA reader at 405 nm. The concentration of reduced and total glutathione was expressed in μmol/mL in 10^5^ cells, respectively.

### 4.10. Transwell Assay for Cell Migration

The cell migration capacity was determined by the ability of cells to pass the pores of the polycarbonate membranes (pore size 8 µm; Greiner Bio-One, Switzerland, St Gallen) at the bottom of the transwell chambers. The migration test was performed according to the protocol described by Chen [44]. The cells were exposed to 1 µg/mL and 10 µg/mL concentrations of treatment FEO, LEO, and SEO for 24 h and 72 h, respectively. The control cells were cultured only in DMEM. After the treatment exposures, all groups of treated cells were trypsinized and placed in the upper chambers at a density of 100,000 cells/well in 500 µL of DMEM with 10% FBS. The lower chambers of the control cells contained 750 µL of DMEM supplemented with 10% FBS, whereas the lower chambers with treated cells were filled with 1 µg/mL and 10 µg/mL concentration of all three treatments. After 6 h of incubation at 37 °C, the cells from the upper surface of the filter were completely removed with gentle swabbing. The remaining migrated cells were fixed for 20 min at room temperature in 4% paraformaldehyde and stained with 0.1% crystal violet in 200 mM 2-(*N*-morpholino) ethanesulfonic acid (pH 6.0) for 10 min; 10% acetic acid dissolved the dye and the absorbance was measured at 595 nm. The migration index was calculated as the ratio of absorbance of the treated samples divided by the absorbance of the nontreated control cell value and multiplied by 100 to give the percentage.

### 4.11. Statistical Analysis

All experiments were performed in triplicate for all the used methods. All data were evaluated using IBM-SPSS 23 software for Windows (SPSS Inc., Chicago, IL, USA). The data were presented as a mean ± standard error (S.E.M). The statistical significance was determined using the paired samples *t*-test. The level of statistical significance was set at *p* < 0.05.

## 5. Conclusions

This study presents a detailed examination of the chemical composition of essential oils obtained from the flowers, leaves, and stems of *T. balsamita*. The obtained results suggest that this plant belongs to the carvone chemotype. The differences between the composition of flowers, leaves, and stems are reflected in the distinct amounts of β-thujone, trans-dihydrocarvone, and cis-carveol, which were more abundant in the EOs from the flowers, 1,8-cineole and α-thujone, with the higher amounts in the leaves EO, and β-bisabolene, which was most abundant in EO obtained from steams.

Antimicrobial testing showed that these EOs have strong activity toward tested bacterial strains. The antimicrobial effects of the vapor phase in the treatment with LEO revealed generally moderate effects on the same bacterial strains. The overall conclusion is that these EOs can be potentially used as natural antimicrobial agents in the pharmaceutical and food industries.

According to their chemical composition, the essential oils are expected to exert an inhibitory effect on cell proliferation/viability, which has been demonstrated in our study. All three essential oils showed antiproliferative effects in MDA-MB-231 and MDA-MB-468 while not significantly affecting the viability of the human lung fibroblast cell line MRC-5, indicating their favorable biocompatibility. In addition, the tested oils induced the considerable reduction in O_2_^•−^ levels, while augmenting the non-enzymatic antioxidative potential of both tested human breast cancer cell lines, which could be one of the major mechanisms of the elevated concentration of nitric oxide recorded in the study. The tested EOs have also exerted antimigratory effects on both tested breast cancer cells and these EOs could exhibit significant antimetastatic properties. The examined EOs show desirable outcomes regarding certain features of the breast cancer cells important for tumor development, which indicates their considerable potential against tumor progression and metastasis. These data indicate that these oils could be interesting and promising agents for further investigations of the signaling pathways of their actions, with the aim of advancing the present antitumor chemotherapeutic strategies.

## Figures and Tables

**Figure 1 plants-11-03474-f001:**
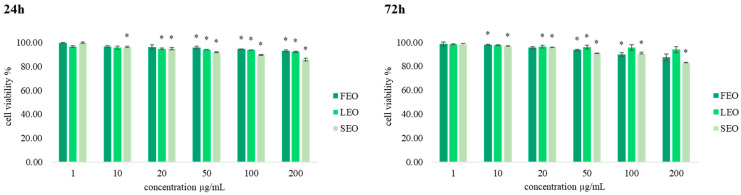
The effects of six concentrations of FEO, LEO, and SEO on MRC-5 cell viability after 24 h and 72 h of treatment. Results are presented as the mean of three independent experiments ± standard error; * *p* < 0.05 relative to control.

**Figure 2 plants-11-03474-f002:**
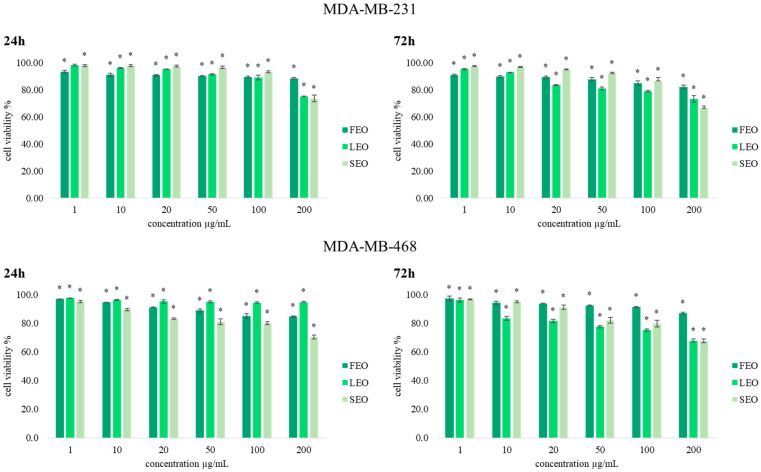
The effects of six concentrations of FEO, LEO, and SEO on MDA-MB-231 and MDA-MB-468 cell viability after 24 h and 72 h of treatment. Results are presented as the mean of three independent experiments ± standard error; * *p* < 0.05 relative to control.

**Figure 3 plants-11-03474-f003:**
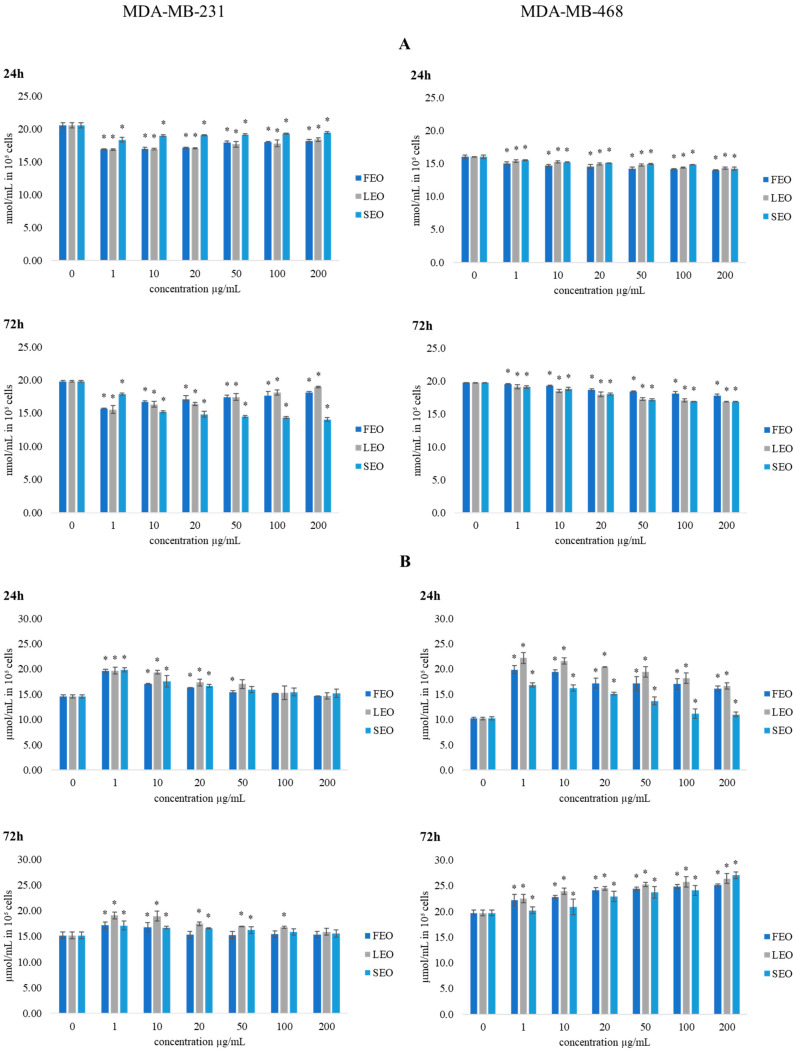
The effects of six concentrations of FEO, LEO, and SEO on the concentration of (**A**) O_2_^•−^ and (**B**) NO_2_^−^ in MDA-MB-231 and MDA-MB-468 cells after 24 and 72 h of treatment. Results are presented as the mean of three independent experiments ± standard error; * *p* < 0.05 relative to control.

**Figure 4 plants-11-03474-f004:**
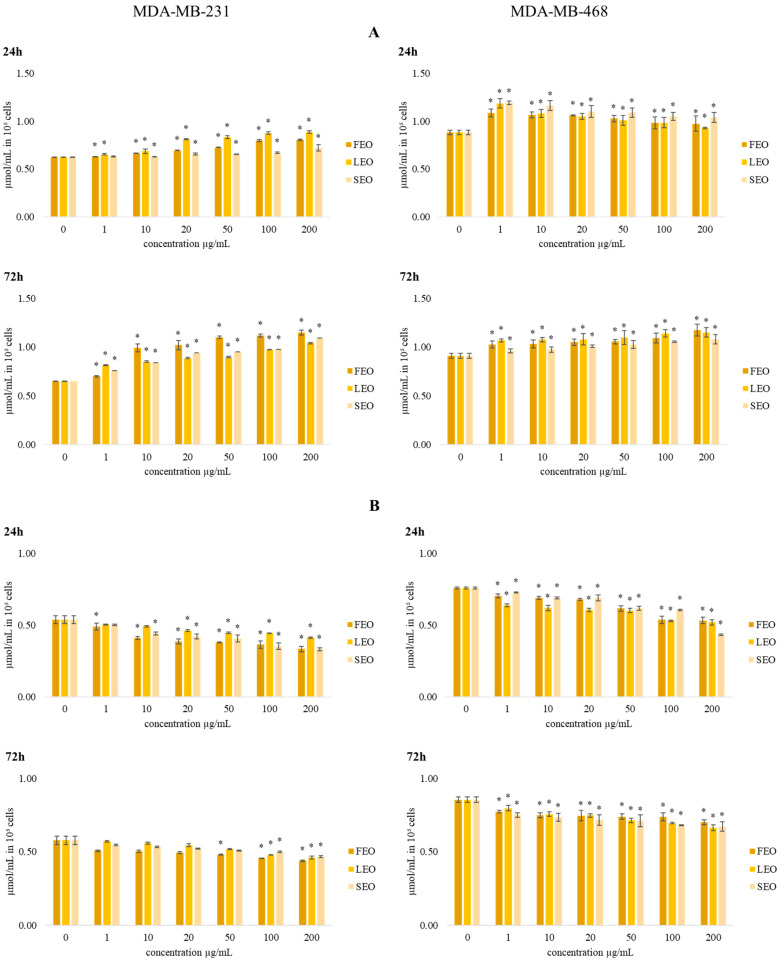
The effects of six concentrations of FEO, LEO, and SEO on the concentration of (**A**) total glutathione and (**B**) reduced glutathione in MDA-MB-231 and MDA-MB-468 cells after 24 and 72 h of treatment. Results are presented as the mean of three independent experiments ± standard error; * *p* < 0.05 relative to control.

**Figure 5 plants-11-03474-f005:**
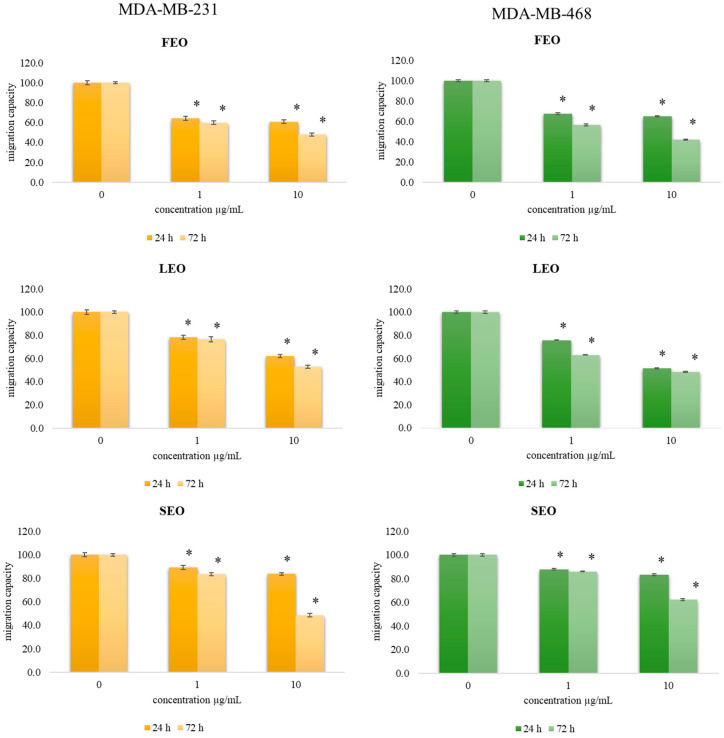
Effect of exposure to investigated FEO, LEO, and SEO on migration index of MDA-MB-231 and MDA-MB-468 cells. The cells were treated at concentrations of 1 µg/mL and 10 µg/mL during 24 h and 72 h exposure compared to nontreated control cells. Results are presented as the mean of three independent experiments ± standard error; * *p* < 0.05 relative to control.

**Table 1 plants-11-03474-t001:** Chemical composition of essential oils from FEO, LEO, and SEO of *T. balsamita*, with corresponding literature and calculated retention indices (RI).

No	Compound ^b^	FEO	LEO	SEO	RI (Lit.)	RI ^a^ (Calc.)
		%				
1	3-methyl-2-butenal	tr ^c^	/^d^	/	784	785
2	1-octene	tr	/	/	791	789
3	*n*-hexanal	0.1	tr	tr	801	800
4	2-methyl-butanoic acid	tr	tr	/	841	845
5	(*Z*)-salvene	tr	0.1	tr	856	851
6	(3*Z*)-hexen-1-ol	tr	tr	/	859	856
7	(*E*)-salvene	/	tr	/	866	862
8	(2*E*)-hexen-1-ol	tr	tr	/	862	865
9	*n*-hexanol	/	tr	tr	870	869
10	2-*n*-butyl furan	tr	tr	/	887	885
11	2-heptanone	tr	/	/	892	891
12	2-heptanol	tr	/	/	896	894
13	*n*-heptanal	/	tr	/	902	903
14	(2*E*,4*E*)-hexadienal	/	tr	/	909	911
15	tricyclene	tr	tr	/	926	926
16	α-thujene	tr	tr	/	930	929
17	allyl isovalerate	tr	tr	/	938	935
18	α-pinene	tr	tr	tr	939	938
19	camphene	0.1	0.2	tr	954	955
20	(2*E*)-heptenal	tr	tr	tr	954	960
21	benzaldehyde	tr	/	tr	960	967
22	1-heptanol	tr	/	/	966	971
23	sabinene	tr	tr	tr	975	977
24	β-pinene	tr	tr	0.2	979	982
25	2-pentylfuran	tr	/	/	988	991
26	butyl butanoate	0.3	0.4	0.4	994	992
27	1,3,5-trimethylbenzene	tr	tr	/	995	996
28	isobutyl-(2*E*)-butenoate	tr	/	/	995	1002
29	ethyl hexanoate	0.4	/	tr	998	1007
30	1-octanal	tr	/	/	998	1011
31	(2*E*,4*E*)-heptadienal	tr	tr	/	1007	1013
32	α-terpinene	tr	tr	tr	1017	1020
33	*p*-cimene	0.5	0.5	0.2	1024	1028
34	limonene	0.4	0.2	0.2	1029	1033
35	1,8-cineole	2	5.9	1.8	1031	1037
36	benzeneacetaldehyde	tr	tr	tr	1042	1047
37	γ-terpinene	0.1	tr	tr	1059	1061
38	α-terpinolene	tr	tr	tr	1088	1087
39	dehydro-linalool	tr	/	tr	1090	1089
40	dehydro-*p*-cymene	tr	tr	tr	1091	1092
41	methyl benzoate	tr	/	/	1090	1096
42	linalool	0.2	/	tr	1096	1100
43	2-methylbutyl 2-methylbutanoate	tr	0.2	tr	1100	1103
44	nonanal	tr	tr	tr	1100	1106
45	α-thujone	1	11.4	7.8	1102	1110
46	β-thujone	6.4	1.5	1.2	1114	1115
47	trans-*p*-menth-2,8-dien-1-ol	1.6	1.5	1.4	1122	1125
48	cis-*p*-menth-2,8-dien-1-ol	1.3	1.11	1	1137	1140
49	trans-pinocarveol	0.9	0.9	0.7	1139	1146
50	trans-verbenol	0.2	0.6	0.5	1144	1150
51	*p*-menth-3-en-8-ol	0.2	/	tr	1150	1155
52	sabina ketone	0.2	0.2	tr	1159	1162
53	pinocarvone	0.9	0.9	0.8	1164	1167
54	borneol	1	0.8	0.6	1169	1177
55	isopinocamphone	tr	tr	tr	1175	1180
56	4-terpinenol	0.4	0.2	0.2	1177	1184
57	*p*-methylacetophenone	tr	tr	0.2	1182	1187
58	trans-*p*-mentha-1(7),8-dien-2-ol	3.5	3.6	3.4	1189	1191
59	myrtenal	tr	0.7	0.5	1195	1198
60	trans-dihydrocarvone	7.7	0.7	1.0	1200	1201
61	trans-carveol	1.5	1.3	1.0	1216	1202
62	2-prenyl cyclopentanone	0.6	1.2	0.5	1227	1214
63	neoiso-dihydro carveol	0.6	0.5	0.4	1228	1220
64	cis-carveol	2.2	0.5	1.3	1229	1223
65	cis-*p*-mentha-1(7),8-dien-2-ol	2.8	2.7	2.5	1230	1235
66	carvone	54.2	52.1	47.7	1243	1250
67	cis-carvone oxide	tr	/	/	1263	1269
68	trans-carvone oxide	0.6	1.1	0.6	1276	1276
69	*p*-cymen-7-ol	tr	/	/	1290	1291
70	*p*-mentha-1,8-dien-7-ol	tr	/	/	1295	1299
71	(2*E*,4*E*)-decadienal	/	/	tr	1316	1316
72	trans-carvyl acetate	0.3	tr	/	1342	1335
73	α-cubebene	tr	tr	tr	1351	1351
74	*p*-eugenol	tr	tr	tr	1359	1355
75	cis-carvyl acetate	0.2	tr	tr	1367	1362
76	piperitenone oxide	/	tr	/	1368	1364
77	α-copaene	tr	tr	tr	1376	1379
78	(*Z*)-β-damascenone	tr	tr	tr	1387	1381
79	(*E*)-caryophyllene	tr	tr	0.2	1419	1421
80	epi-bicyclosesquiphellandrene	0.2	tr	0.1	1493	1473
81	α-muurolene	tr	tr	/	1500	1497
82	β-bisabolene	1.2	4.0	7.7	1505	1507
83	δ-amorphene	tr	tr	0.6	1512	1515
84	δ-cadinene	1.0	0.5	1.0	1523	1518
85	cis-calamenene	0.3	tr	0.3	1529	1522
86	(*E*)-γ-bisabolene	/	/	0.6	1531	1531
87	trans-cadina-1,4-diene	tr	/	tr	1534	1533
88	α-calacorene	tr	tr	0.4	1545	1542
89	spathulenol	0.2	0.3	0.7	1578	1578
90	caryophyllene oxide	0.3	0.2	0.5	1583	1583
91	β-copaen-4-α-ol	0.5	0.5	0.7	1590	1588
92	ledol	tr	0.2	0.8	1602	1604
93	β-cedrene epoxide	/	/	0.8	1622	1629
94	10-epi-γ-eudesmol	0.4	0.3	/	1623	1629
95	gossonorol	/	/	0.3	1637	1640
96	τ-muurolol	0.4	0.4	1.2	1642	1645
97	β-eudesmol	tr	/	0.3	1650	1657
98	α-cadinol	0.5	0.6	0.7	1654	1661
99	8-cedren-13-ol	/	/	0.2	1689	1677
100	(*Z*)-α-trans-bergamotol	/	/	0.8	1690	1687
101	2-heptadecanone	tr	/	tr	1900	1899
102	methyl hexadecanoate	tr	/	/	1921	1924
103	hexadecanoic acid	/	/	0.1	1960	1957
total		97.4	98.0	94.1		

^a^ Values of retention indices on HP-5MS column; ^b^ identified compounds; ^c^ tr—compounds identified in amounts less than 0.1%; ^d^/- not identified.

**Table 2 plants-11-03474-t002:** Percentage composition of each class of identified compounds.

Class of Compounds	FEO	LEO	SEO
	%		
**nonterpenic compounds**			
hydrocarbons	tr ^a^	0.1	tr
alcohols	tr	tr	tr
aldehydes	0.1	tr	tr
ketones	tr	/ ^b^	tr
esters	0.7	0.6	0.4
acids	tr	tr	0.1
aromatic compounds	tr	tr	0.2
heterocyclic compounds (furan)	tr	tr	/
*subtotal*	**0.8**	**0.7**	**0.7**
**monoterpenes**			
*monoterpene hydrocarbons*	1.1	0.9	0.6
*summ*	1.1	0.9	0.6
*oxygenated monoterpenes*			
monoterpene alcohols	16.4	13.7	13.0
monoterpene aldehydes	tr	0.7	0.5
monoterpene ketones	71.6	69.1	59.6
monoterpene esters	0.5	tr	tr
monoterpene ethers	2.0	5.9	1.8
*summ*	90.5	89.4	74.9
*subtotal*	**91.6**	**90.3**	**75.5**
**phenylpropanoids**	tr	tr	tr
*subtotal*	**tr**	**tr**	**tr**
**sesquiterpenes**			
*sesquiterpene hydrocarbons*	2.7	4.5	10.9
*summ*	2.7	4.5	10.9
*oxygenated sesquiterpenes*			
sesquiterpene alcohols	2.0	2.3	5.7
sesquiterpene epoxides	0.3	0.2	1.3
*summ*	2.3	2.5	7.0
*subtotal*	**5.0**	**7.0**	**17.9**
**total**	**97.4**	**98.0**	**94.1**

^a^ tr—compounds identified in amounts less than 0.1%; ^b^/- not identified.

**Table 3 plants-11-03474-t003:** Number of identified volatile compounds in investigated EOs samples.

Class of Compounds	FEO	LEO	SEO
	Number of Compounds		
**nonterpenic compounds**			
hydrocarbons	2	2	1
alcohols	4	3	1
aldehydes	6	6	4
ketones	2	/^a^	1
esters	6	3	3
acids	1	1	1
aromatic compounds	5	3	3
heterocyclic compounds (furan)	2	1	/
*subtotal*	**28**	**19**	**14**
**monoterpenes**			
*monoterpene hydrocarbons*	12	12	10
*summ*	12	12	10
*oxygenated monoterpenes*			
monoterpene alcohols	16	11	14
monoterpene aldehydes	1	1	1
monoterpene ketones	11	11	10
monoterpene esters	2	2	1
monoterpene ethers	1	1	1
*summ*	31	26	27
*subtotal*	**43**	**38**	**37**
**phenylpropanoids**	1	1	1
*subtotal*	**1**	**1**	**1**
**sesquiterpenes**			
*sesquiterpene hydrocarbons*	11	10	11
*summ*	11	10	11
*oxygenated sesquiterpenes*			
sesquiterpene alcohols	7	6	9
sesquiterpene epoxides	1	1	2
*summ*	8	7	11
*subtotal*	**19**	**17**	**22**
**total**	**91**	**75**	**74**

^a^/- not identified.

**Table 4 plants-11-03474-t004:** Antibacterial activity of tested EOs from *T. balsamita* and standard antibiotics.

Bacteria	MIC50 (MIC90) μg/mL				
	FEO	LEO	SEO	MEM	VAN
*Gram-positive*					
*L. monocytogenes*	0.246 ± 0.003 (0.436 ± 0.001)	0.246 ± 0.003 (0.436 ± 0.001)	0.246 ± 0.003 (0.436 ± 0.001)	0.63 ± 0.04 (0.85 ± 0.05)	0.63 ± 0.04 (0.85 ± 0.05)
*S. aureus*	0.53 ± 0.05 (0.61 ± 0.05)	0.016 ± 0.002 (0.022 ± 0.001)	0.009 ± 0.002 (0.011 ± 0.01)	0.33 ± 0.03 (0.56 ± 0.05)	0.021 ± 0.003 (0.032 ± 0.004)
*S. pneumoniae*	0.53 ± 0.05 (0.61 ± 0.05)	0.13 ± 0.02 (0.18 ± 0.02)	0.13 ± 0.02 (0.18 ± 0.02)	0.33 ± 0.03 (0.56 ± 0.05)	0.33 ± 0.03 (0.56 ± 0.05)
*Gram-negative*					
*E. coli*	0.246 ± 0.003 (0.436 ± 0.001)	0.246 ± 0.003 (0.436 ± 0.001)	0.53 ± 0.05 (0.61 ± 0.05)	0.33 ± 0.03 (0.56 ± 0.05)	0.021 ± 0.003 (0.032 ± 0.004)
*H. influenzae*	0.063 ± 0.004 (0.088 ± 0.002)	0.063 ± 0.004 (0.088 ± 0.002)	0.036 ± 0.002 (0.054 ± 0.004)	0.041 ± 0.001 (0.063 ± 0.002)	0.33 ± 0.03 (0.56 ± 0.05)
*Y. enterocolitica*	0.016 ± 0.002 (0.022 ± 0.003)	0.036 ± 0.002 (0.054 ± 0.004)	0.036 ± 0.002 (0.054 ± 0.004)	0.041 ± 0.002 (0.063± 0.002)	0.072 ± 0.003 (0.091 ± 0.003)

MEM—meropenem; VAN—vancomycin.

**Table 5 plants-11-03474-t005:** Antibacterial activity in situ of the vapor phase of leaves EO (LEO) from *T. balsamita* against six bacterial strains growing on selected food models.

**Food Model**	**Bacteria**	**Bacterial Growth Inhibition (%)**		
		**Concentration of LEO in μL/L**		
		3.9	7.8	15.6
Apple	*Gram-positive*			
	*L. monocytogenes*	−5.12 ± 0.43	25.30 ± 1.06	−24.46 ± 2.06
	*S. aureus*	−54.29 ± 2.62	26.02 ± 2.09	44.23 ± 0.99
	*S. pneumoniae*	35.25 ± 1.07	12.01 ± 0.49	15.83 ± 0.83
	*Gram-negative*			
	*E. coli*	−31.40 ± 1.07	−12.89 ± 1.50	34.35 ±0.98
	*H. influenzae*	23.45 ± 0.90	−6.17 ± 0.25	−37.34 ± 0.89
	*Y. enterocolitica*	6.73 ± 1.20	−29.76 ± 3.78	−6.34 ± 0.89
Pear	*Gram-positive*			
	*L. monocytogenes*	36.32 ± 1.12	23.32 ± 0.95	14.45 ± 0.86
	*S. aureus*	−63.19 ± 1.04	−34.61 ± 1.57	13.26 ± 0.84
	*S. pneumoniae*	25.24 ± 1.04	5.36 ± 0.12	2.76 ± 0.42
	*Gram-negative*			
	*E. coli*	6.69 ± 0.39	23.85 ± 2.03	12.57 ± 0.48
	*H. influenzae*	35.31 ± 0.92	−9.28 ± 0.55	17.43 ± 0.06
	*Y. enterocolitica*	6.73 ± 1.20	9.65 ± 0.50	34.22 ± 2.02
Carrot	*Gram-positive*			
	*L. monocytogenes*	−5.04 ± 0.81	−26.08 ± 3.21	−56.10 ± 2.39
	*S. aureus*	19.32 ± 0.95	14.05 ± 1.50	−24.45 ± 0.86
	*S. pneumoniae*	5.32 ± 0.81	9.9 ± 0.45	17.43 ± 1.00
	*Gram-negative*			
	*E. coli*	16.34 ± 1.85	25.06 ± 0.45	6.73 ± 1.20
	*H. influenzae*	47.87 ± 1.35	27.46 ± 1.06	−45.25 ± 1.73
	*Y. enterocolitica*	13.31 ± 0.95	−36.25 ± 0.90	−23.24 ± 1.09
White radish	*Gram-positive*			
	*L. monocytogenes*	−35.36 ± 1.01	−5.96 ± 0.52	−9.05 ± 6.54
	*S. aureus*	−17.09 ± 1.48	−4.97 ± 0.25	17.67 ± 0.48
	*S. pneumoniae*	15.68 ± 0.57	8.19 ± 0.68	35.29 ± 1.06
	*Gram-negative*			
	*E. coli*	26.12 ± 1.50	45.22 ± 1.03	87.35 ± 1.97
	*H. influenzae*	35.28 ± 0.94	74.23 ± 1.04	23.54 ± 1.49
	*Y. enterocolitica*	87.09 ± 1.48	4.26 ± 1.08	16.34 ± 1.85

Mean ± standard deviation. The negative values indicate a probacterial activity of the essential oil against the growth of bacterial strains.

## Data Availability

Not applicable.

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
