# Peer review of "Chemical Composition and Biological Activity of Tanacetum balsamita Essential Oils Obtained from Different Plant Organs"

_plants, 2022, doi:10.3390/plants11243474_

Round 1

Reviewer 1 Report

The manuscript, after having eviscerated the composition of the essential oils from T. balsamita, deals with its biological activity expressed as antimicrobial and anticancer properties. The study has been well designed and conducted and the conclusions obtained are consistent with the procedures adopted. However, before its acceptance, some points should be addressed

Abstract

Abstract should be reduced in length.

Introduction

Page 3, line 100: please delete the word proapoptotic, as MTT test does not provide information about apoptosis.

Results

·         Page 5, lines 114-115: why ninety-one…, seventy-five…, seventy-four and not 91, 75, 74?

·         The results presented on pages 9 and 10 (sub-headings 2.3.1 to 2.3.4) can be merged together within 2.3 In situ antibacterial analysis on a food model, also because they are shown in a single table (Table 5).

·         Also the results presented on pages 11-12 (sub-headings 2.5.1 e 2.5.2) can be merged together within the single sub-heading 2.5.1 Determination of superoxide anion radical (NBT assay) and nitrites (Griess Assay). Once again results are shown in one figure (Figure 2, A and B). Of course, sub-heading 2.5.3 will become 2.5.2

Discussion

I would not emphasize too much the antiproliferative capacity of these essential oils since values of cell viability, although significantly different from the control, never falls below 60% in both cell lines, values besides ​​obtained at a high extract concentration (200 µg/ml). Or at least I would mention that in the discussion.

Author Response

Dear Ms. Emeline Chen

Thank you for the review of our manuscript entitled “Chemical composition and biological activity of Tanacetum balsamita essential oils obtained from different plant organs. The Authors are very grateful to the Reviewer for their valuable comments. We would like to thank the Reviewer for the time devoted for constructive and important comments to improve our paper. All possible changes in the manuscript have been introduced.

Yours sincerely,

Nenad Vuković

Reviewer #1

The manuscript, after having eviscerated the composition of the essential oils from T. balsamita, deals with its biological activity expressed as antimicrobial and anticancer properties. The study has been well designed and conducted and the conclusions obtained are consistent with the procedures adopted. However, before its acceptance, some points should be addressed

The Authors are very grateful to the Reviewer for their valuable comments. We would like to thank the Reviewer for the time devoted for constructive and important comments to improve our paper.

Point 1: Abstract.

Abstract should be reduced in length.

Response: The abstract was shortened. Please see lines 17-36.

Point 2: Introduction

Page 3, line 100: please delete the word proapoptotic, as MTT test does not provide information about apoptosis.

Response: We have deleted the word proapoptotic. Please see line 97.

Point 3: Results

Page 5, lines 114-115: why ninety-one…, seventy-five…, seventy-four and not 91, 75, 74?

Response: We have changed words into numbers. Please see line 111-112.

The results presented on pages 9 and 10 (sub-headings 2.3.1 to 2.3.4) can be merged together within 2.3 In situ antibacterial analysis on a food model, also because they are shown in a single table (Table 5).

Response: We have merged all subheadings.

Also the results presented on pages 11-12 (sub-headings 2.5.1 e 2.5.2) can be merged together within the single sub-heading 2.5.1 Determination of superoxide anion radical (NBT assay) and nitrites (Griess Assay). Once again results are shown in one figure (Figure 2, A and B). Of course, sub-heading 2.5.3 will become 2.5.2

Response: We have merged all subheadings.

Point 4: Discussion

I would not emphasize too much the antiproliferative capacity of these essential oils since values of cell viability, although significantly different from the control, never falls below 60% in both cell lines, values besides ​​obtained at a high extract concentration (200 µg/ml). Or at least I would mention that in the discussion.

Response: Thank you for your observation. We have added the explanation within the Discussion section. Please see lines 383-385.

Reviewer 2 Report

Lines 341-449

It should be stated whether or not the differences in the composition of the essential oils extracted from different parts of the costmary plants, i.e. flower (FEO), leaf (LEO) and stem (SEO), are significant and whether or not they correlate with the antibacterial activity

Lines 614-619

Additionally, the results obtained should be also be analyzed using tests that would allow the results for the control combinations to be compared with the experimental groups (the other variants of each experiment) and homogeneous groups to be determined. The individual variants in the conducted experiments cannot be compared with each other based on the statistical calculations performed in the paper.

Lines 620-646

A series of further tests should be performed, and it is the results of these tests that should be used to support the conclusion that essential oils can be potentialy used as natural antimicrobal agents in the pharmaceutical and food industries.

Author Response

Dear Ms. Emeline Chen

Thank you for the review of our manuscript entitled “Chemical composition and biological activity of Tanacetum balsamita essential oils obtained from different plant organs. The Authors are very grateful to the Reviewer for their valuable comments. We would like to thank the Reviewer for the time devoted for constructive and important comments to improve our paper. All possible changes in the manuscript have been introduced.

Yours sincerely,

Nenad Vuković

Reviewer #2

Thank you very much for your suggestions.

Point 1: Lines 341-449; It should be stated whether or not the differences in the composition of the essential oils extracted from different parts of the costmary plants, i.e. flower (FEO), leaf (LEO) and stem (SEO), are significant and whether or not they correlate with the antibacterial activity.

Response: We clearly presented results (Part- Volatile composition of examined essential oils; 101-168) of the chemical composition of volatiles isolated from different plant parts. These results were also properly discussed and connected with other literature data in Part Discussion (first and second paragraph; 322-344). On the other hand, in the third paragraph (355-361) we clearly connect compositions of essential oils with antibacterial activity. In this paragraph, we mentioned carvone as the main constituent of all investigated essential oils. Also, in this part we wrote that carvone can be responsible for the strong activity. Again, from a scientific point, we must consider the synergistic effect of the other components present in minor amounts (as we wrote - `` Obviously, we cannot overlook the synergistic effect of the other components present in minor amounts that may affect the effectiveness of the exhibited antimicrobial effect``).

Point 2: Lines 614-619; Additionally, the results obtained should be also be analyzed using tests that would allow the results for the control combinations to be compared with the experimental groups (the other variants of each experiment) and homogeneous groups to be determined. The individual variants in the conducted experiments cannot be compared with each other based on the statistical calculations performed in the paper.

Response: Thank you for your proposition. The presented results do not represent the experimental group, but only serial doses of treatments applied to the cells. Accordingly, the design of the study and discussion was only based on a screening assessment of antitumor performances of three essential oils compared to non-treated cells. However, in some future studies, it could be informative to compare the in-between effects of the treatments.

Point 3: Lines 620-646; A series of further tests should be performed, and it is the results of these tests that should be used to support the conclusion that essential oils can be potentialy used as natural antimicrobal agents in the pharmaceutical and food industries.

Response: These results were obtained as a part of project research (APVV-20-0058 and 451-03-68/2022-14/200122). Since we are at the beginning of the investigation of the costmary plant (exactly for experiments included in this research) it is better to write `` can be potentially used``. Of course, for practical implementation of these essential oils in the food and pharma industry we will perform several tests related to toxicity. But, for these tests we must obtain special ethical permissions (after that we will go further). 

Reviewer 3 Report

I really appreciate your work. I found it interesting and pretty well exposed. However, I would like to evidence the following point:

1. Fig.1-Pag. 11. There is no 0 point (concentration) or control to use as reference/control in any of the four boxes. Why?

2. You have used two different units to describe the concentration of the essential oil μg/mL and μl/L (in the vapor phase test). Why? In the material and methods for the MIC you stated that you started whut a serial dilution of essential oil from 1 μl/mL, while in the results you have used μg/mL; on the other hand, on cell cell cultivation and treatment the doses ranged from (1 μg/mL to 200 μg/mL). Why did you use different units of measures? It gives some confusion in interpretating the dose/effects in the different contests.  

3. Have you use some solubilizer for essential oil? It does seems not from the materials and methods If not, given the highly lipophilic nature of essential oils,  how you can be sure that the different doses the essential oil are effectively dispersed in the growing/testing media? Have you done some test about that?  

Author Response

Dear Ms. Emeline Chen

Thank you for the review of our manuscript entitled “Chemical composition and biological activity of Tanacetum balsamita essential oils obtained from different plant organs. The Authors are very grateful to the Reviewer for their valuable comments. We would like to thank the Reviewer for the time devoted for constructive and important comments to improve our paper. All possible changes in the manuscript have been introduced.

Yours sincerely,

Nenad Vuković

Reviewer #3

I really appreciate your work. I found it interesting and pretty well exposed. However, I would like to evidence the following point:

The Authors are very grateful to the Reviewer for their valuable comments. We would like to thank the Reviewer for the time devoted for constructive and important comments to improve our paper.

Point 1: 1. Fig.1-Pag. 11. There is no 0 point (concentration) or control to use as reference/control in any of the four boxes. Why?

Response: Thank you for your question. The concept of the Figure 1 does not contain 0 concentration as a control, because the value of 100% of viability is considered the control (non treated) value, and all other treatment values are presented in relation to this control value in Figure 1.

Point 2: You have used two different units to describe the concentration of the essential oil μg/mL and μl/L (in the vapor phase test). Why? In the material and methods for the MIC you stated that you started whut a serial dilution of essential oil from 1 μl/mL, while in the results you have used μg/mL; on the other hand, on cell cell cultivation and treatment the doses ranged from (1 μg/mL to 200 μg/mL). Why did you use different units of measures? It gives some confusion in interpretating the dose/effects in the different contests. 

Response: We had typing mistake, for the calculation of MIC it was necessary to weigh the mass of the oils, we have made that modification is the Materials and methods part, please see the lines 484-487. But for the vapor phase test method does not require oils to be measured in mass.

Point 3: Have you use some solubilizer for essential oil? It does seems not from the materials and methods If not, given the highly lipophilic nature of essential oils,  how you can be sure that the different doses the essential oil are effectively dispersed in the growing/testing media? Have you done some test about that? 

Response: Respected Reviewer, yes. We absolutely agree with this observation. Avoiding insufficient dissolution is easy, and researchers who test biological activities of essential oils use the same and well know methodology. We used the same methodology as in our previous research. The stock solutions were prepared by dissolving the oils in a media/DMSO mixture in a ratio of 9:1 (v/v) and the oils have been dispersed homogenously with no layers formed. All treatment concentrations were obtained by serial dilutions of stock solution, so DMSO concentrations decreased continuously and never exceeded 0.2% in the maximal applied concentration.